# ON DIFFICULTIES OF PROBABILITY DISTILLATION

## ABSTRACT

Probability distillation has recently been of interest to deep learning practitioners as it presents a practical solution for sampling from autoregressive models for deployment in real-time applications. We identify a pathological optimization issue with the commonly adopted stochastic minimization of the (reverse) KL divergence, owing to skewed gradient distribution due to curse of dimensionality. We also explore alternative principles for distillation, and show that one can achieve qualitatively better results than with KL minimization.

## 1 INTRODUCTION

Deep autoregressive models are currently among the best choices for unsupervised density modeling tasks (Van den Oord et al., 2016b;a) However, due to the factorization induced by such models on the data space by the chain rule of probability, sampling from such models requires $\mathcal{O}(T)$ sequential computation steps, where $T$ is the dimension/sequence-length of the data. This makes sampling slow and inefficient for most practical purposes. A recent solution to alleviating this bottleneck was proposed by Van den Oord et al. (2018), who show how to distill an autoregressive *WaveNet* model (Van den Oord et al., 2016a) into a student network, which is significantly faster to sample from, and therefore much more suitable for deployment in real-time applications. In this paper, we identify a fundamental issue with the approach taken by Van den Oord et al. (2018), and provide alternative views and principles for distilling the sampling process of a teacher model.

The solution proposed by Van den Oord et al. (2018) is to minimize the reverse Kullback-Leibler divergence (KL) between the student and the teacher distribution. Generally, this relies on two essential components: (1) the gradient signal from the teacher network (the WaveNet model) and (2) invertibility of the student network (the Parallel WaveNet model). This allows samples drawn from the student to be evaluated under the likelihood of both models, and the student can then be updated via stochastic backpropagation from the teacher. The student is implemented with an inverse autoregressive transformation (Kingma et al., 2016) that, unlike an autoregressive model, admits fast sampling at the cost of slow likelihood estimation.

In theory, it seems that minimizing the KL should be sufficient for distilling a teacher into a student. However, in practice, it has been necessary to implement additional heuristics in order to achieve samples with similar levels of realism as that of the teacher (Van den Oord et al., 2018; Ping et al., 2018). In particular, a *power loss* that biases the power across frequency bands of the student sampler to match that of human speech patterns has been crucial in recent work for generating synthesized speech without the student collapsing to unrealistic 'whispering'.

The main contribution of this work is to identify that the reverse KL is ill-suited for the task of distillation, since the teacher distribution typically possesses a low-rank nature for high-dimensional structured data, which renders the effective gradient signal sparse. We explore some possible alternatives for distillation, by recasting the problem of distilling a generative model as learning a transformation of probability density from a prior space to the data space, connecting decoder-based generative models with probability density distillation. We present experimental results which demonstrate that it is possible to learn qualitatively better samplers than distillation with reverse KL.

## 2 AUTOREGRESSIVE MODELS

Given a joint probability distribution $p(x)$, where $x$ denotes a $T$-dimensional vector, one can factorize the distribution according to the *chain rule of probability*, for arbitrary ordering of dimensions:

$$p(x) = p(x_1)p(x_2|x_1)...p(x_T|x_1,...,x_{T-1}) = \prod_{t=1}^{T} p(x_t|x_{1:t-1})$$

For the tabular case, i.e. when $x_t$ can take $V$ different possible values, the size of the table is $\mathcal{O}(V^T)$. When the event set of $x_t$ is uncountable, the joint density is not even tractable. This motivates the use of a parametric model to compress the conditional probability $p(x_t|x_{1:t-1})$, where one has $p_\theta(x) = \prod_{t=1}^{T} p_\theta(x_t|x_{1:t-1})$. This is referred to as an *autoregressive model*. Parameters are usually shared across the dimensions of $x$, since $T$ might not stay constant, for example in the case of recurrent neural networks or convolutional neural networks, which have been empirically demonstrated to possess good inductive biases for tasks involving images (Van den Oord et al., 2016b) and speech data (Van den Oord et al., 2016a). However, sampling is sequential, requiring $T$ passes, which is why state-of-the-art autoregressive models have been slow to sample from, which makes them impractical for tasks requiring sampling, such as speech generation. This has motivated the work of Van den Oord et al. (2018), who propose *probability density distillation* to learn a student network with a structure that allows for parallel sampling, by distilling a state-of-the-art autoregressive teacher into it.

## 3 PROBABILITY DISTILLATION WITH NORMALIZING FLOWS

Van den Oord et al. (2018) propose to distill the probability distribution parameterized by a WaveNet model (denoted by $\mathcal{T}$, which stands for the teacher network) by minimizing its reverse KL with a student network (denoted by $\mathcal{S}$):

$$D_{\mathrm{KL}}(p_\mathcal{S} \,||\, p_\mathcal{T}) = \mathbb{E}_{x \sim p_\mathcal{S}}[\log p_\mathcal{S}(x) - \log p_\mathcal{T}(x)] \tag{1}$$

The idea is to leverage recent advances in *change of variable* models (also known as *normalizing flows*) to parallelize the computation of the sampling process. First, the student distribution is constructed by transforming an initial distribution $p_\mathcal{S}(z)$ (e.g. normal or uniform distribution) in a way such that each dimension $x_t$ in the output $x$ depends only on up to $t$ preceding variables (according to a chosen ordering) in the input $z$:

$$z_t \sim p_\mathcal{S}(z),$$

$$x_t \leftarrow g_t(z_t; \pi_t(z_1, ..., z_{t-1})),$$

where $g_t$ is an invertible map between $z_t$ and $x_t$. Unlike the sampling process of an autoregressive model, where one needs to accumulate all $x_{1:t-1}$ to sample $x_t$ from $p_\mathcal{T}(x_t|x_{1:t-1})$, which scales $\mathcal{O}(t)$, the transformations $g_t$ can be carried out independently of $t$, allowing for $\mathcal{O}(1)$ time sampling. Second, the entropy term of $p_\mathcal{S}$ can be estimated using the change of variable formula:

$$\mathbb{E}_{x \sim p_\mathcal{S}(x)}[\log p_\mathcal{S}(x)] = \mathbb{E}_{z \sim p_\mathcal{S}}\left[\log p_\mathcal{S}(z)\left|\frac{\partial g(z)}{\partial z}\right|^{-1}\right] \tag{2}$$

where $g$ is the overall transformation $g(z)_t \doteq g_t(z_t; z_{<t})$. Furthermore, owing to the partial dependency of $g_t$, $g$ has a triangular Jacobian matrix, reducing the computation of the log-determinant of Jacobian in (2) to linear time:

$$\left|\frac{\partial g(z)}{\partial z}\right| = \prod_t \frac{\mathrm{d}g_t(z_t; z_{<t})}{\mathrm{d}z_t}$$

Finally, when the teacher network has a tractable explicit density, one can evaluate the likelihood of samples drawn from $p_\mathcal{S}$ under $p_\mathcal{T}$ efficiently (in the case of autoregressive models such as WaveNets, one can use teacher forcing to compute log-likelihood in parallel).

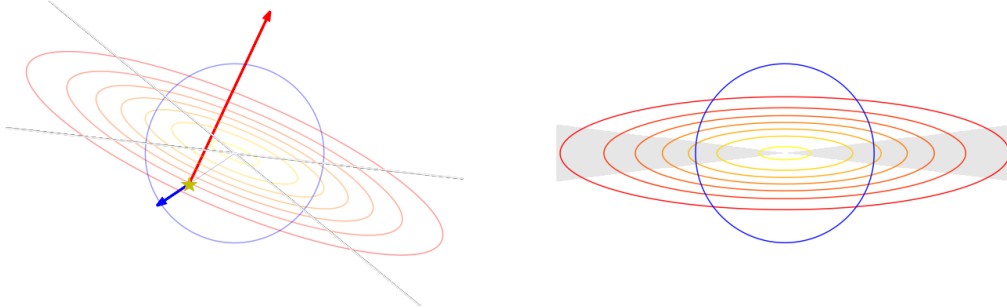

(a) Gradients from teacher (red) and student (blue).  (b) Rotated by eigenvectors of $\Sigma_\mathcal{T}$.

Figure 1: When distilling the teacher distribution (red) with a student distribution (blue) by minimizing the KL divergence, the student receives two counteracting gradient signals from the two corresponding likelihood terms. The probability of receiving a signal to push out the student distribution to fill the teacher is proportional to the relative amount of space occupied by the shaded area, which vanishes when the covariance matrix of the teacher has trivial eigenvalues.

## 3.1 CURSE OF DIMENSIONALITY

In this section, we explain the failure mode of distillation with reverse KL that we have identified. Intuitively, for each sample $x$ drawn from $p_\mathcal{S}$, the negative gradient of the integrand in Equation (1) with respect to $x$, is made up of two counteracting factors: one that pushes $x$ away from the mode of the student density $p_\mathcal{S}$ and one that pulls $x$ towards the mode of the teacher density $p_\mathcal{T}$. These two counteracting terms form the "error" component of the *path derivative* (Roeder et al., 2017) when the student is updated. (see Appendix C for more details). Assume the student density $p_\mathcal{S}$ lies within a certain family $\mathcal{Q}$. Ideally, if $p_\mathcal{T} \in \mathcal{Q}$, the solution to the minimization problem in (1) would be $p_\mathcal{S} = p_\mathcal{T}$. However, this is not trivial in practice when an oracle solver is not accessible. We empirically find out that training with reverse KL consists of two stages:

(i) $p_\mathcal{S}$ starts to fit to the mode of $p_\mathcal{T}$ (Figure 2a)and

(ii) $p_\mathcal{S}$ gradually expands from the mode of $p_\mathcal{T}$ to fit the shape of the distribution.

Stage (i) is fast due to the well-known zero forcing property of the reverse KL. Turner & Sahani (2011) (see Fig 1.3) shows that $p_\mathcal{S}$ tends to be more concentrated when independence assumption is made. We show that even when $\mathcal{Q}$ contains $p_\mathcal{T}$, stochastic optimization can result in slow convergence of stage (ii) and thus a more concentrated, suboptimal $p_\mathcal{S}$. This is because the gradient signal via the path derivative can be increasingly unlikely to be effective when the dimensionality in $x$ grows and when $p_\mathcal{T}$ is highly structured. This makes it harder for $p_\mathcal{S}$ to expand its probability mass along the high density manifold under $p_\mathcal{T}$. We illustrate this as follows.

Consider the case when both the student and teacher are multivariate normal centered at the origin: assume without loss of generality[1] that $p_\mathcal{S} = \mathcal{N}(0, \mathbb{I})$ and $p_\mathcal{T} = \mathcal{N}(0, \Sigma_\mathcal{T})$ (both centered at 0 assuming stage (i) has been completed), where $\mathbb{I}$ is an $n$-by-$n$ identity matrix and $\Sigma_\mathcal{T} \in S^n_{++}$ is a positive definite matrix. The following proposition establishes the connection between the eigenvalues of the covariance matrix $\Sigma_\mathcal{T}$ and the probability of getting a positive signal that pushes $x$ away from the origin.

**Proposition 1.** *Let $p_\mathcal{S} = \mathcal{N}(0, \mathbb{I})$ and $p_\mathcal{T}(0, \Sigma_\mathcal{T})$. Draw $x \sim p_\mathcal{S}$. Let $A_U$ be the surface area of the unit sphere: $U = \{x : \|x\|_2 = 1\}$ and $A_{U_{\cap \rho}}$ be the surface area of $\{x \in U : \sum_i \rho_i x_i^2 > 0\}$. Then the probability of $\nabla_x[\log p_\mathcal{T} - \log q_\mathcal{S}]$ pointing away from the origin along $x$ is given by*

$$\frac{A_{U_{\cap \rho}}}{A_U} \tag{3}$$

---

[1]When the covariance matrix of the student distribution $\Sigma_\mathcal{S}$ is not an identity matrix, one can transform both $p_\mathcal{S}$ and $p_\mathcal{T}$ via the change of variable: $x' = U^{-\top}x$ where $\Sigma_\mathcal{S} = U^\top U$ is the Cholesky decomposition of the covariance matrix; such that $p_\mathcal{S}(x')$ is standardized, $p_\mathcal{T}(x')$ has a "relative" covariance (due to the rotation under $U^{-\top}$), and our analysis carries on.

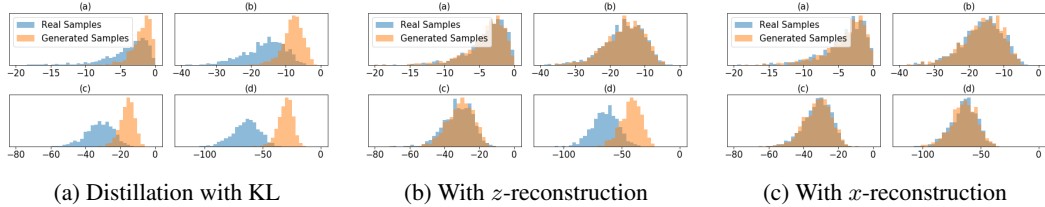

(a) Distillation with KL       (b) With $z$-reconstruction       (c) With $x$-reconstruction

Figure 2: We distill a Gaussian teacher with a Gaussian student. $x$-axis: likelihood under the teacher; $y$-axis: count of samples drawn from the teacher (real samples) and the learned student (generated samples). (a-d) in the subfigures correspond to $\{4, 16, 32, 64\}-$ dimensional multivariate Gaussians.

*where $\rho_i = 1 - \frac{1}{d_i^2}$ and $d_i^2$ is the $i$-th eigenvalue of the covariance matrix.*

*Proof.* Let $g_x \doteq \nabla_x(\log p_{\mathcal{T}}(x) - \log p_{\mathcal{S}}(x))$ be the gradient. Since both $p_{\mathcal{S}}$ and $p_{\mathcal{T}}$ are Gaussian distributions centered at the origin, the gradient when projected onto $x$, either points towards or away from the origin, depending on whether $x^\top g_x < 0$ or $x^\top g_x > 0$. By definition, we have $g_x = -\Sigma_{\mathcal{T}}^{-1} x + x$. Let $\Sigma_{\mathcal{T}} = \Lambda D \Lambda^{-1}$ be the eigen-decomposition of the covariance, where $D_{ii} = d_i^2$ is the $i$-th eigenvalue and the columns of $\Lambda$ are the eigenvectors. Due to the rotational invariance and uniformity of the density on the level set $\{x : \|x\|_2 = r\}$ for any $r > 0$ of the standard normal $p_{\mathcal{S}}$,

$$\mathbb{P}\left\{x^\top g_x > 0\right\} = \mathbb{P}\left\{x^\top x - x^\top \Lambda D^{-1}\Lambda^{-1}x > 0\right\} = \mathbb{P}\left\{\sum_i(1 - \frac{1}{d_i^2})x_i^2 > 0\right\} = \frac{A_{U \cap \rho}}{A_U}$$

$\square$

What the proposition implies is that the chances of receiving a gradient signal that points outward depend on the eigenvalues of the covariance matrix of the teacher: the greater the number of eigenvalues that are smaller than 1 (more ill-conditioned), the lower the chances. Consider Figure 1a for example, where the red contour plot and blue contour plot represent the density of $p_{\mathcal{T}}$ and $p_{\mathcal{S}}$, respectively. For a random sample drawn from $p_{\mathcal{S}}$, marked by the yellow star, the gradients of $\log p_{\mathcal{T}}$ and $-\log p_{\mathcal{S}}$ with respect to it are represented by the red arrow and blue arrow. The net $g_x$ here can be decomposed into two parts: one that is perpendicular to $x$, $g_{x,\perp}$, and one that is parallel with $x$, $g_{x,\|}$. In this example, $x$ and $g_{x,\|}$ point towards the opposite direction, meaning the back-propagated signal would draw $x$ towards the mode of $p_{\mathcal{T}}$. On average, the chances of getting a stochastic gradient signal that push the points away from the mode is the percentage of the area of the unit sphere intersecting with the hypercone, represented by the shaded area in Figure 1b. In practice, such a condition coefficient can be very small, as it is well known that a high dimensional distribution over structured data is effectively low-rank. In fact, an almost-sure convergence result of the smallest eigenvalue of a random Wishart matrix scaled by $1/T$ was proven by Silverstein et al. (1985) to be effectively zero with large enough $T$. The Marchenko-Pastur Law describes a more general asymptotic distribution of the eigenvalues (Marchenko & Pastur, 1967).

### 3.2 EMPIRICAL DEMONSTRATION

We showed above that with increasing dimensionality, and for a structured teacher, there is a diminishing probability of gradient signals that can push the student to expand around the mode of the teacher. It seems intuitive that the distilled density of the student will therefore be collapsed around the mode of the teacher density. We validate this hypothesis in the following experiment.

We take both $p_{\mathcal{T}}$ and $p_{\mathcal{S}}$ to be multivariate Gaussian distributions, with the sampling process defined as $x \leftarrow \mu + R \cdot z$ where $\mu \in \mathbb{R}^T$, $R \in \mathbb{R}^{T \times T}$ and $z \sim \mathcal{N}(0, \mathbb{I})$. We randomly initialize each element of $R$ for $\mathcal{T}$ independently according to the standard Gaussian, set $\mu = [2, ..., 2]^\top$ to be a vector of $T$ 2's, and fix them while training $\mathcal{S}$ to distill $\mathcal{T}$. For $T \in [4, 16, 32, 64]$, training proceeds as follows: we sample $x$ from the student, estimate $\log p_{\mathcal{S}}(x)$ using the change of variable formula, and evaluate $x$ under $\log p_{\mathcal{T}}$. We use a minibatch size of $64$ and learning rate of $0.005$ with the Adam

optimizer (Kingma & Ba, 2014), and make 5000 updates. For evaluation, we draw 1000 samples from both $p_\mathcal{T}$ and $p_\mathcal{S}$, and display the empirical distribution of $\log p_\mathcal{T}(x)$ in Figure 2a.

First, with increasing number of dimensions, we observe that $p_\mathcal{S}$ does indeed concentrate more on the high density region of $p_\mathcal{T}$. This suggests that the unbalanced gradient signal poses an optimization problem for distillation of higher dimensional structured distributions; getting a sample that gets pushed away along the thin manifold under the teacher density is very unlikely, which is consistent with Proposition 1.

Second, we also observe that the log-likelihood of the teacher samples deviate from 0 as dimensionality grows. In fact, assuming $x_t$ is sampled i.i.d., the $l_2$ norm of $\bar{x} \doteq \frac{x}{\sqrt{T}}$ would almost surely converge. This is a phenomenon known as the *concentration of measure*. To see this,

$$\|\bar{x}\|_2^2 = \bar{x}^\top \bar{x} = \sum_{t=1}^T \bar{x}_t^2 = \sum_{t=1}^T \left(\frac{x_t}{\sqrt{T}}\right)^2 = \frac{1}{T}\sum_{t=1}^T x_t^2 \longrightarrow \mathbb{E}[x_t^2] \quad \text{a.s. as } T \to \infty$$

The concentration is due to the compromise between density and volume of space (which vanishes exponentially as dimensionality grows). The consequence is that when one samples from a high dimensional Gaussian, the norm of the sample can be well described by its expected value, which means one is effectively sampling from the shell of the Gaussian ball. This suggests that the use of KL would result in a mismatch of certain important statistics (such as norm of the samples, which is a perceivable feature in images and audio frames) even when $p_\mathcal{S}$ is fairly close to $p_\mathcal{T}$.

Finally, in the above study, we only identify this optimization difficulty in the convex setup. However, it is also well known that the reverse KL tends to be mode-seeking (see Figure 3b,3c for example), and is not well-suited for learning multimodal densities (Turner & Sahani, 2011; Huang et al., 2018b).

## 4 PROBABILITY DISTILLATION WITH INVERSE MATCHING

In this section, we discuss possible alternatives for distilling a teacher. We assume there exists an invertible mapping from a prior space $\mathcal{Z}$ to the data space $\mathcal{X}$, such that one can trivially sample from a prior distribution $z \sim p_\mathcal{T}(z)$ and pass the sample through this invertible map such that the sample is distributed according to $p_\mathcal{T}(x)$. For Gaussian conditional autoregressive models, for example, one would sequentially pass scalar standard Gaussian noise $z_t$ through the following recursive function $x_t = \mu_t(x_{1:t-1}) + \sigma_t(x_{1:t-1}) \cdot z_t$. For notational convenience, we denote the "inverse" of this transformation by $\mathcal{T} : \mathcal{X} \to \mathcal{Z}$, as this is the inverse autoregressive transformation that can be parallelized. The goal is to learn the sampling transformation, i.e. $\mathcal{T}^{-1}$. Similarly, we define the forward pass of the student as the mapping $\mathcal{S} : \mathcal{Z} \to \mathcal{X}$.

Ideally, since the transformation is deterministic, it's most natural to simply minimize the prediction loss according to some distance metric $d(\mathcal{T}^{-1}(z), \mathcal{S}(z))$, where $z \sim p_{\mathcal{T},z}$. When this loss function equals zero almost everywhere, passing the prior sample through $\mathcal{S}$ would induce an identical distribution as $p_\mathcal{T}$. We refer to this setup as **distillation with oracle prediction**. However, preparing such a dataset of $\mathcal{T}^{-1}(z)$ samples would typically be time-consuming. We present the following two alternatives.

**1. Distillation with $z$-reconstruction.** We consider minimizing $d(z, \mathcal{T} \circ \mathcal{S}(z))$, which is a reconstruction loss and the student network and teacher network are viewed as the encoder and decoder, respectively. In this case, since $\mathcal{T}$ is invertible and fixed, the only functional form of $\mathcal{S}$ that gives zero reconstruction would be $\mathcal{T}^{-1}$, which means the random variable $\mathcal{S}(Z)$ should also be distributed according to $p_\mathcal{T}$. In fact, minimizing the $z$-reconstruction loss corresponds to a parametric distance induced by the teacher network. Define $d_\mathcal{T}(a, b) \doteq d(\mathcal{T}(a), \mathcal{T}(b))$, where $d$ is a distance metric. Then

$$d_\mathcal{T}(\mathcal{T}^{-1}(z), \mathcal{S}(z)) = d(\mathcal{T} \circ \mathcal{T}^{-1}(z), \mathcal{T} \circ \mathcal{S}(z)) = d(z, \mathcal{T} \circ \mathcal{S}(z)).$$

Interestingly, $d_\mathcal{T}$ is also a metric:

**Proposition 2.** $d_\mathcal{T}$ *is a metric if and only if $\mathcal{T}$ is injective.*

*Proof.* Trivially, positive-definiteness and symmetry are inherited from $d$ if and only if $\mathcal{T}$ is an injection. To see that subadditivity is also preserved, for some $a$, $b$ and $c$, let $\mathcal{T}_a = \mathcal{T}(a)$, $\mathcal{T}_b = \mathcal{T}(b)$

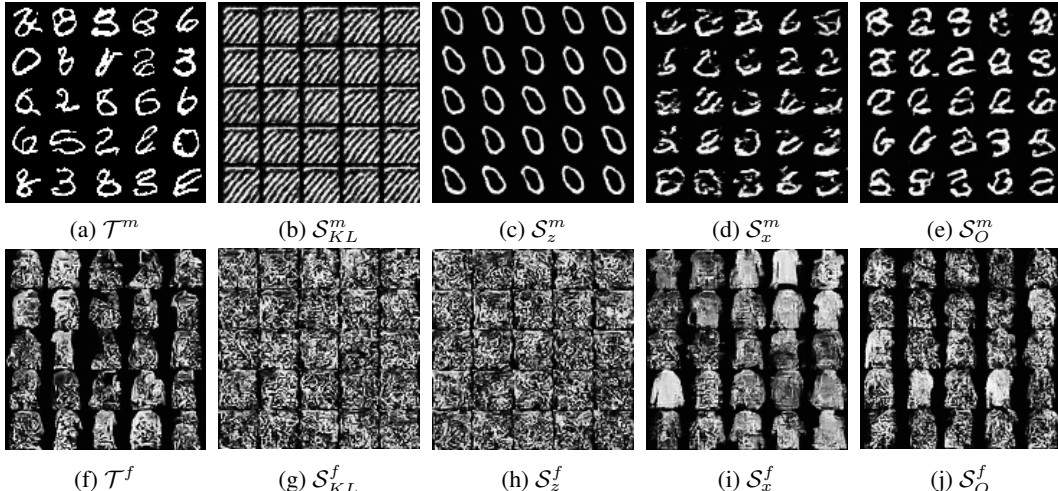

| (a) $\mathcal{T}^m$ | (b) $\mathcal{S}^m_{KL}$ | (c) $\mathcal{S}^m_z$ | (d) $\mathcal{S}^m_x$ | (e) $\mathcal{S}^m_O$ |
|---|---|---|---|---|
| (f) $\mathcal{T}^f$ | (g) $\mathcal{S}^f_{KL}$ | (h) $\mathcal{S}^f_z$ | (i) $\mathcal{S}^f_x$ | (j) $\mathcal{S}^f_O$ |

Figure 3: Density distillation of teacher models trained on MNIST (first row) and Fashion-MNIST(second row). *Column 1*: samples from teacher network $\mathcal{T}$. *Column 2*: samples from student trained with the KL loss $\mathcal{S}_{KL}$. *Column 3*: samples from student trained with the $z$-reconstruction loss $\mathcal{S}_z$. *Column 4*: samples from student trained with the $x$-reconstruction loss $\mathcal{S}_x$. *Column 5*: samples from student trained with the $x$-reconstruction loss where $x$ is sampled from the teacher $\mathcal{S}_O$.

and $\mathcal{T}_c = \mathcal{T}(c)$. Since $d(\mathcal{T}_a, \mathcal{T}_b) \leq d(\mathcal{T}_a, \mathcal{T}_c) + d(\mathcal{T}_b, \mathcal{T}_c)$, due to the subadditivity of $d$, for any $\mathcal{T}_a$, $\mathcal{T}_b$ and $\mathcal{T}_c$, we have $d_\mathcal{T}(a, b) \leq d_\mathcal{T}(a, c) + d_\mathcal{T}(b, c)$ for any $a$, $b$ and $c$. $\qquad\square$

This means z-reconstruction loss behaves like a distance between $\mathcal{T}^{-1}(z)$ and $\mathcal{S}(z)$. So when z-reconstruction is minimized, it implies $\mathcal{S}$ gets closer to $\mathcal{T}^{-1}$ in the sense of the induced metric $d_\mathcal{T}$.

**2. Distillation with $x$-reconstruction.** Finally, we consider minimizing the reconstruction loss $d(x, \mathcal{S} \circ \mathcal{T}(x))$, where $x \sim p_\mathcal{D}$, the (empirical) data distribution, taking the teacher network as the encoder, and the student network as the decoder. When the teacher density coincides with the underlying data distribution, this would be equivalent to training with oracle prediction, as $\mathcal{T}(X)$ would be distributed according to $p_\mathcal{T}(z)$. This is a reasonable assumption when $p_\mathcal{T}$ approximates $p_\mathcal{D}$ well, and this is in fact true as $p_\mathcal{T}$ is usually trained with Maximum likelihood under $p_\mathcal{D}$.

Now we revisit the two essential components required for distillation with reverse KL:

(1) **Invertibility**: None of the three training criteria we explored involves estimating the entropy of $p_\mathcal{S}$, so in principle, we do not require invertibility of the student. In fact the entropy of $p_\mathcal{S}$ is implicitly maximized since $\mathcal{T}$ is bijective. To prevent degenerate $p_\mathcal{S}$, one simply needs to avoid using hidden units of dimensionality smaller than the input size without skip connectivity, which compresses the noise.

(2) **Differentiability**: For distillation with oracle prediction and $x$-reconstruction, we only require the translation between $\mathcal{X}$ and $\mathcal{Z}$, via $\mathcal{T}$ and $\mathcal{T}^{-1}$, which is readily accessible for many standard distributions, e.g. linear map between Gaussians (both $\mathcal{T}$ and $\mathcal{T}^{-1}$), logistic-linear map from mixture of logistics to uniform ($\mathcal{T}$ only, but sampling is achievable by sampling the mixture component first), and neural transformation (Huang et al., 2018a) ($\mathcal{T}$ only). We also note that it is possible to recover uniform density from discrete data, by injecting noise proportional to the probability per class to break ties when passing the data through the cumulative sum of the probability (CDF).

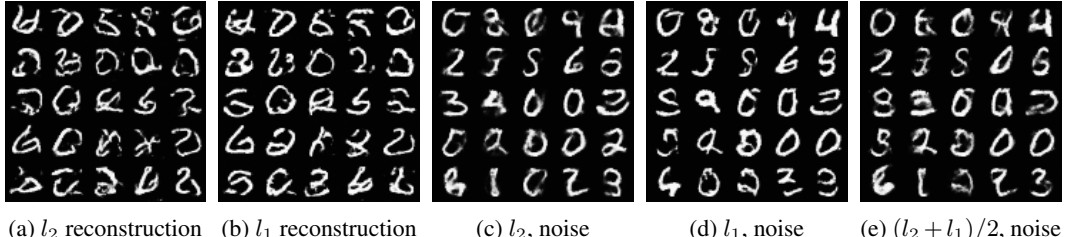

(a) $l_2$ reconstruction    (b) $l_1$ reconstruction    (c) $l_2$, noise    (d) $l_1$, noise    (e) $(l_2 + l_1)/2$, noise

Figure 4: Experiments with a ResNet student using (a) $l_2$ $x$-reconstruction loss, (b) $l_1$ $x$-reconstruction loss, (c) adding $N(0, 0.5)$ noise to the encodings $z$ and using $l_2$ $x$-reconstruction loss, (d) adding $N(0, 0.5)$ noise to $z$ and using $l_1$ $x$-reconstruction loss, and (e) adding $N(0, 0.5)$ noise to $z$ and using a mixed loss $l_2 + l_1$. In general, we observe that adding noise significantly improves sample quality, and training with $l_1$ losses lead to sharper samples.

## 4.1 EXPERIMENTS

### 4.1.1 LINEAR MODEL WITH INCREASING DIMENSIONALITY

We replicate the experiment in Section 3.2 with $z$-reconstruction and $x$-reconstruction loss (equivalent to oracle prediction in this case). Mapping $\mathcal{T}$ from $\mathcal{X}$ to $\mathcal{S}$ is simply inverse of the sampling transformation. We observe that both models outperform distillation with reverse KL, since likelihood of samples (under the teacher) drawn from them follow more closely the shape of the empirical distribution of likelihood of samples drawn from the teacher. It is worth noting that $z$-reconstruction also starts to fail with an increasing number of dimensions, suggesting that it might be subject to poor gradient signal if the transformation $\mathcal{T}$ is more complex; we elaborate more on this in the next section. On the other hand $x$-reconstruction is quite robust to dimensionality (and potentially complexity) of the underlying distribution.

### 4.1.2 DISTILLATION WITH DIFFERENT LOSSES

In this section, we distill PixelCNN++ (Salimans et al., 2017) teacher networks trained on the MNIST handwritten digits dataset (LeCun et al., 1998) and the Fashion-MNIST dataset (Xiao et al., 2017). We trained the teacher model for 100 epochs and distilled it into the student with another 100 epochs of updates, using minibatch size of 64, learning rate of 0.0005 for the Adam optimizer with a decay rate of 0.95 per epoch, 3 ResNet blocks per downsampling and upsampling convolution, 32 hidden channels, and a single Gaussian conditional. The data is preprocessed with uniform noise between pixel values and rescaled using the logit function. We use $l_2$ loss for the reconstruction and prediction methods.

First, we observe that when trained with the reverse-KL loss, the students collapse on undesirable modes. As shown in (Figure 3b), the inductive bias of the causal convolution leads to higher density of the samples with striped textures. When trained with the $z$-reconstruction loss, the MNIST student samples all collapse to the same digit. Interestingly, when we visualize the corresponding $\mathcal{T}^{-1}(z)$ as we slightly perturb the norm of $z$ (see Appendix A), we observe that the digits abruptly change identity. This suggests that when moving onto a different sublevel set of norm in $z$-space, the corresponding $x$ jumps from one digit manifold to another, and the direction of $z$ does not preserve digit identity. This might explain why the student collapses to a digit: this is due to the bad local minimum that corresponds to relatively low reconstruction cost in the $z$-space.

Next, we observe that the student trained with $x$-reconstruction loss (Figure 3d) does not have good quality samples while the reconstructions are visually perfect. We hypothesize this is due to the well-known problem of mismatch between the empirical distribution of the encodings $\mathcal{T}(x)$ and the prior distribution $p_{\mathcal{S}}(z)$ of training decoder based generative models (Kingma et al., 2016). We contrast this with a student trained on oracle predictions (Figure 3e) and observe that the latter's samples match the teacher's samples better.

Finally, we see that the samples from the teacher trained on Fashion-MNIST with the $x$-reconstruction loss (Figure 3i) have a smoother texture than the one trained with oracle samples

(Figure 3j), which again, are perceptually closer to samples from the teacher. We elaborate more on this discrepancy in the next section.

### 4.1.3 LEARNING TO DISTILL AND LEARNING TO GENERATE

Since we are not constrained in our modeling choice for the student, we experiment with a ResNet student which is trained to directly map an encoded datapoint from the teacher back to the datapoint. The ResNet is deep enough so that the receptive field at the output is sufficient to span all of the encoded $z$, so that far-away influence is still exploitable.

Using the $l_1$ reconstruction cost leads to sharper samples from the student (contrast 4a with 4b). It appears to us that the $l_2$ loss tends to maintain global details, while the $l_1$ loss can sometimes sacrifice global coherence for local structure, potentially due to the sparsity induced by it. In 4e, we use an average of both losses in an attempt to maintain both these characteristics, but we notice evidence of the failings manifesting to some extent as well.

A significant improvement in sample quality occurs upon adding Gaussian noise to the encodings before training the student (contrast 4a,4b with 4c,4d). Our intuition for this is as follows: providing a decoder network with pairs of points in a data space and an encoded (or latent) space would typically result in almost all of $z$-space not being "trained", especially since image data (and therefore the corresponding encodings in $z$-space) usually lies in a low-dimensional manifold. Adding noise enhances the support of the distribution, effectively spreading the distribution of "responsibility" of an encoding to cover more volume, which smoothens the mapping learned by a decoder. When the goal is to sample from a prior, training methods that encourage such $z$-space-filling strategies and smoother mappings improve sample quality, in the vein of decoder-based sampling models such as variational auto-encoders (Kingma & Welling, 2014).

This leads us to an important point about these experiments: since the student is trained on noised (encoded) points from the data distribution, this is no longer purely density distillation. The student no longer aims to reproduce the sampling behavior of the teacher (as in 3e), but rather uses the teacher to provide structural information through its encodings. This information, when "spread out" through noise-injection and used by a student to learn decodings into real data (through a reconstruction penalty or more sophisticated losses) results in a network that can now be considered as a stand-alone generator, with the teacher acting as an inference machine that preserves information in the latent space. This can potentially allow a student to outperform its teacher in terms of sample quality, by enabling the learning of a smoother mapping from $z$-space to data space.

### 4.1.4 NEURAL VOCODER

We compare distillation with $x$-reconstruction and the reverse-KL approach on the *neural vocoder* Sotelo et al. (2017) task for speech synthesis. The neural vocoder is an essential component of many text-to-speech models proposed recently (Wang et al., 2017; Shen et al., 2018; Arik et al., 2017; Ping et al., 2018). We train our teacher to map vocoders (Morise et al., 2016) to raw audio using the SampleRNN (Mehri et al., 2016) model. We model the conditional distribution of the teacher with a unimodal gaussian distribution, making it easier to compute the corresponding $z$. We specifically compare against closed form regularized KL with Gaussian conditionals as proposed in (Ping et al., 2018). The student network has a WaveNet architecture with six flows, and performs sampling as in Parallel WaveNet (Van den Oord et al., 2018). Each flow is a dilated residual block of 10 layers with a convolution kernel width of 2 and 64 output channels. We use our $x$-reconstruction method with $l_1$ as the reconstruction loss. We empirically find that our method results in a student with samples without the characteristic whispering of the reverse-KL trained student. We have uploaded samples for comparison here [2].

## 5 CONCLUSION

In this paper, we investigate problems with distilling an autoregressive generative model under a reverse KL cost between the student and the teacher, where the student can perform efficient parallel generation. Specifically, we show that distillation with the reverse KL can suffer from imbalanced

---

[2] https://soundcloud.com/inverse-matching/sets/samples-for-inverse-matching

gradient signals due to the curse of dimensionality. Further, we explore different alternatives which work qualitatively better when compared with distillation with reverse KL.

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

## A VISUALIZATION OF $\mathcal{T}^{-1}(z)$ WITH PERTURBED NORM OF $z$

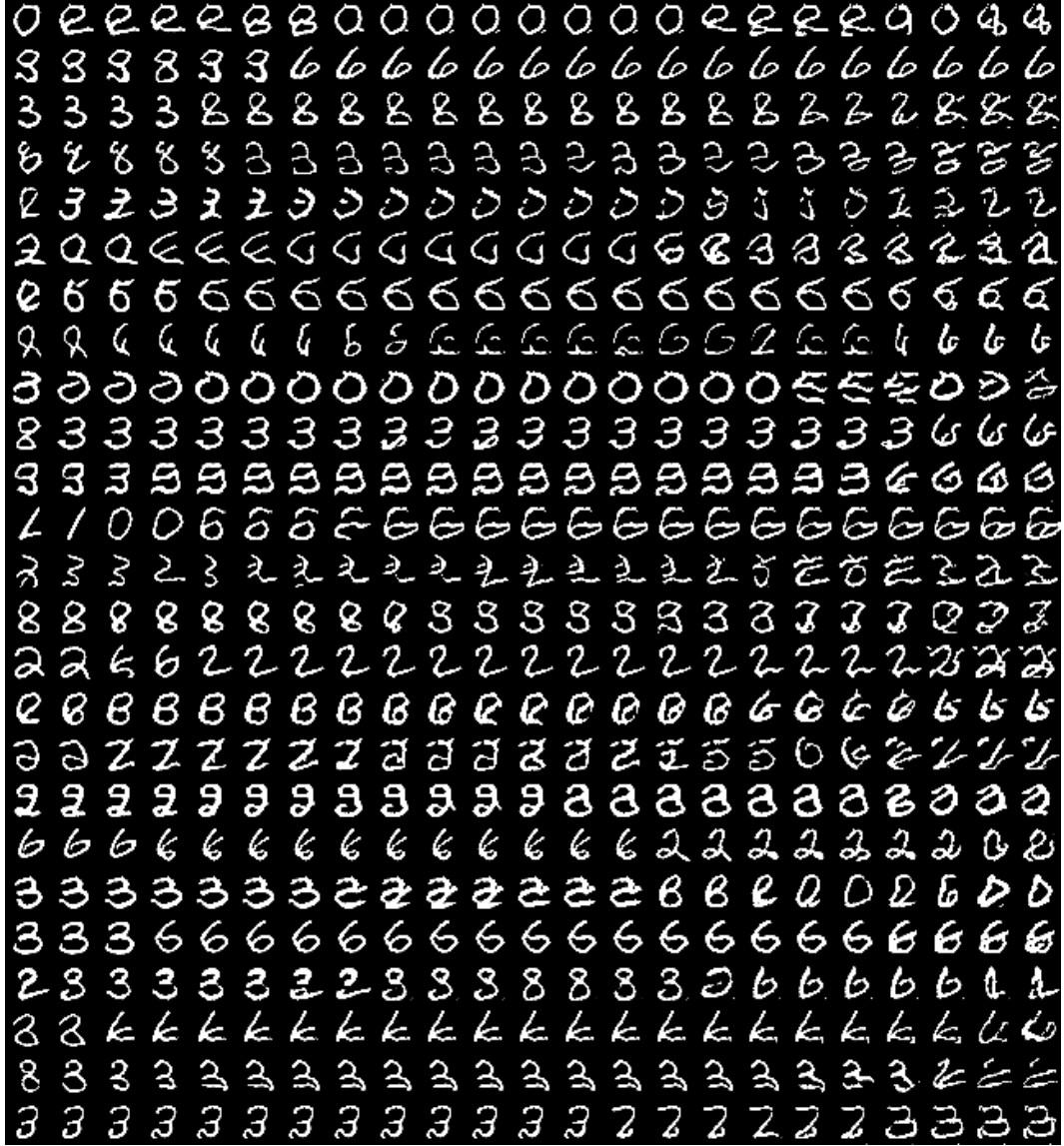

Figure 5: We randomly sample $z$ from $\mathcal{N}(0, \mathbb{I}_{784})$ (for each row) and rescale the vector such that it has norm $r \cdot \sqrt{784}$, where $r \in [0.700, 0.750, 0.800, 0.850, 0.900, 0.920, 0.940, 0.960, 0.980, 0.990, 0.995, 1.000, 1.005, 1.010, 1.020, 1.040, 1.060, 1.080, 1.100, 1.150, 1.200, 1.250, 1.300]$, which correspond to the change along the horizontal axis. We observe that directional information does not preserve digit identity in the data space, and the manifold per digit can be stretched around the origin on different level sets of norm.

## B    ADDITIONAL RESULT ON DISTILLING PIXELCNN TRAINED ON CIFAR-10 WITH X-RECONSTRUCTION (NOISE INJECTION)

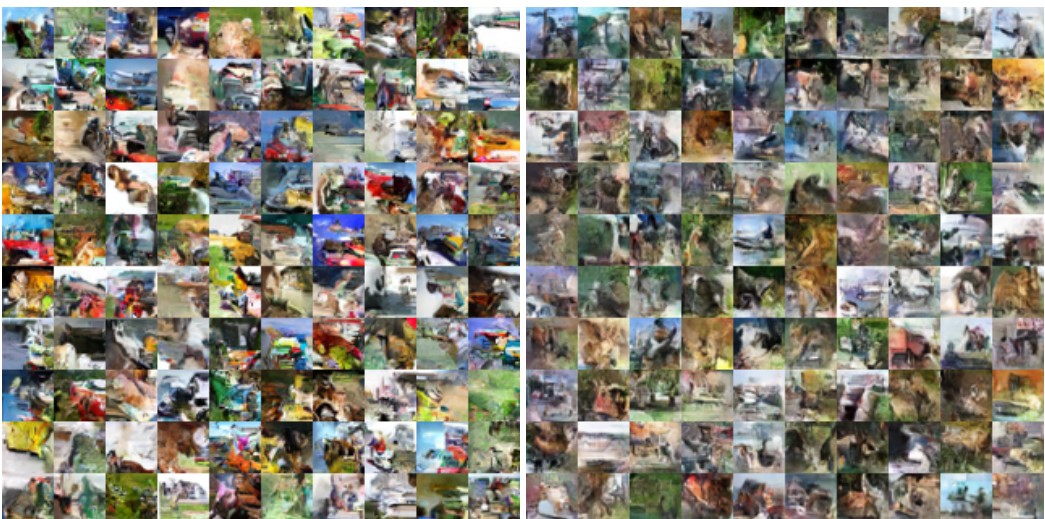

Figure 6: (left) Teacher samples with a PixelCNN on CIFAR-10, (right) Student samples using $x$-reconstruction under the $l_1$ loss, with noise injection.

## C    PATH DERIVATIVE

When $x$ is real-valued and when reparameterization of the sample drawn from $p_{\mathcal{S}}(x)$ allows for separation of a random variable $z \sim p(z)$ independent of the parameters of $p_{\mathcal{S}}$ and a deterministic transformation $x = \mathcal{S}_\phi(z)$, we can decompose the gradient of the KL divergence (1) with the parameters $\phi$ (of $\mathcal{S}$) as (Roeder et al., 2017):

$$
\begin{aligned}
\nabla_\phi \mathbb{E}_{x \sim p_{\mathcal{S}_\phi}(x)} & [\log p_{\mathcal{S}_\phi}(x) - \log p_{\mathcal{T}}(x)] \\
&= \nabla_\phi \mathbb{E}_{z \sim p(z)}[\log p_{\mathcal{S}_\phi}(x) - \log p_{\mathcal{T}}(x)] \\
&= \mathbb{E}_{z \sim p(z)}[\nabla_\phi (\log p_{\mathcal{S}_\phi}(x) - \log p_{\mathcal{T}}(x))]; \qquad (x = \mathcal{S}_\phi(z)) \\
&= \mathbb{E}_{z \sim p(z)}[\underbrace{\nabla_x (\log p_{\mathcal{S}_\phi}(x) - \log p_{\mathcal{T}}(x)) \nabla_\phi \mathcal{S}_\phi(z)}_{\text{path derivative}} + \underbrace{\nabla_\phi \log p_{\mathcal{S}_\phi}(x)}_{\text{score function}}]
\end{aligned}
$$

where the last equality is the total derivative because the term $p_{\mathcal{S}_\phi}(x)$ depends on $\phi$ through both the sample being evaluated, $x = S_\phi(z)$, and the evaluating log-likelihood function, $\log p_{\mathcal{S}_\phi}$.

The score function is in expectation zero, since

$$
\int p_{\mathcal{S}_\phi}(x) \nabla_\phi \log p_{\mathcal{S}_\phi}(x) dx = \int p_{\mathcal{S}_\phi}(x) \frac{\nabla_\phi p_{\mathcal{S}_\phi}(x)}{p_{\mathcal{S}_\phi}(x)} dx = \nabla_\phi \int p_{\mathcal{S}_\phi}(x) dx = \nabla_\phi 1 = 0,
$$

so it can be thought of as a control variate: an unbiased term used to reduce variance in gradient estimate. The path derivative term measures the dependency on the parameters $\phi$ through the reparameterized sample $x = \mathcal{S}_\phi(z)$. As a result, the gradient direction wrt the sample $x$ directly affects how the parameters of the distribution will be updated to change the shape of $p_{\mathcal{S}_\phi}$.

