# OpenReview forum: "On Difficulties of Probability Distillation"
_ICLR.cc/2019/Conference_

### Official Review · AnonReviewer3 · 2018-11-02
**Interesting idea & fair results**

**Rating:** 5
**Confidence:** 4

**Review:**

This paper analyzes the limitation of probability density distillation with reverse KL divergence, and proposes two practical methods for probability distillation.

Detailed comments:

1) Typo: should be WaveNet, not Wavenet.

2) In Proposition 1. $c_i$ should be $\rho_i$.

3) One may explain “path derivative” with more details. Also, I am really confused by Proposition 1 and its underlying implication. Given p_s and p_t are centered at the origin, isn’t p_s(x) already the optimal if it’s just a unit Gaussian. Why do we need a derivative pointing away from the origin? At least, one need parameterize p_s as N(0, \phi)?

4) In section 3.2, “set $\mu = [2, 2]^T$”? Isn’t $\mu$ a T dimensional vector?

5) A lot of important details are missing in neural vocoder experiment. For x-reconstruction, do you use L1 or L2 loss?  For student model, do you use Gaussian IAF with WaveNet architecture as in ClariNet, or Logistic IAF as in Parallel WaveNet? Following this question, do you compute KLD in closed-form? Do you use the regularization term introduced in ClariNet? Student with KL loss and power loss outperforms x-reconstruction. Did you try x-reconstruction along with power loss?

Pros:
Certainly, there are some interesting ideas in this paper.

Cons:
The experiment results are not good enough. The paper is poorly written. A lot of important details are missing.

However, I would like to raise my rating to 6, if these comments can be properly addressed.

---

> ### Author Response · Authors · 2018-11-22
> **Response to AnonReviewer3**
>
> Thank you for the constructive feedback! We address the concerns below:
>
> 1. Thanks for spotting the typo! We’ve fixed it.
>
> 2. We used c initially to express the formula generally, for any vector c. This is a stylistic choice, but we agree that it might be confusing for some readers, and for the sake of readability, we’ve changed the c to rho.
>
> 3. We've updated Sec3.1 in the hope to better motivate and explain the analysis. Here's some clarification.
>     a. We’ve included a derivation and explanation of the path derivative in the appendix; in short, the path derivative consists of the gradient direction wrt the sample x, which directly affects how the parameters of the distribution will be updated to change the shape of p_S.
>     b. p_S is optimal when p_S=p_T (iff KL=0). In practice, what we found is that when the algorithm tries to minimize KL, p_S tends to fit to the mode of p_T and the mass is overly concentrated. This “mode collapse” problem, as manifested in practical issues such as the whispering characteristic present in the reverse-KL trained Parallel Wavenet, is indeed the key issue motivating our study!
>     c. p_T being a high dimensional gaussian is simply a “model of the problem” to demonstrate how stochastic optimization can be inefficient due to the “unbalanced” gradient distribution. Assume now p_S is sharper than p_T, the gradient in expectation should point to a direction that expands the probability mass of p_S (along the high density valley under p_T). Our analysis suggests that even when this is true in expectation, one might have exponentially low probability to sample a gradient with the required expansion signal: e.g. more contractive gradients with smaller magnitude and less expansive gradients with larger magnitude.
>
> 4. Thanks for spotting the typo! We’ve fixed it to set $\mu=[2,...,2]^\top$ to be a vector of $T$ $2$'s
>
> 5. We’ve updated our submission to include more details, which we also present here: For the vocoder experiment, we used the L1 loss, Gaussian IAF as in ClariNet. Each flow consists of 10 residual dilated convolution blocks (these are the standard blocks used in Parallel WaveNet) with kernel-width 2 and 64 output channels. We compute the regularized KL in closed form (for the baseline experiments with KL), using vocoder as input.

---

### Official Review · AnonReviewer2 · 2018-11-02
**Convincing paper about of a potentially not-too-widespread technical issue.**

**Rating:** 7
**Confidence:** 2

**Review:**

The paper studies the problem of distilling a student probabilistic model (that
is easy to sample from) from a complex teacher model (for which sampling is
slow).  The authors identify a technical issue with a recent distillation
technique, namely that positive gradient signals become increasingly unlikely
as the dimensionality of the teacher model increases.  They then propose two
alternative technique that sidestep this issue.

The topic is definitely relevant.  The paper focus on a single method for
probability distillation, which limits the significance of the contribution.

The paper is very well written and well structured.  Section 4 is may be a bit
too dense for the uninitiated; it may make sense to clarify that calT and calS
refer to the teacher and student models---it is only obvious while reading this
section for the second time around.

All contributions seem novel.  The fact that the (reverse) KL can lead to bad
models is known; the issue identified in this paper, however, seems novel.

I could not spot any major flaws with the paper.

The evaluation is satisfactory.  The issue of KL-based training is very clear,
as is the advantage of the encoder-decoder alternatives.

I especially appreciated the link between distillation and encoder-decoder
architectures.

Detailed comments:

1 - How widespread is the issue identified in this paper?  In other words, is
reverse KL realistically used in applications other than probability
distillation?

2 - It is unclear to me why Proposition 2 is important.  This should be
explicitly stated.

3 - It would make sense to add a forward pointer to Figure 3c in Section 3.1,
to provide another example of mode-seeking.

---

> ### Author Response · Authors · 2018-11-22
> **Response to AnonReviewer2**
>
> Thank you for your positive review and feedback! We address the comments below:
>
> 1. Reverse KL minimization has been used in many other contexts other than the recent application of distilling an autoregressive model which we focus on in this paper. Most common applications have been in areas such as variational inference [1], variational continual learning [2], energy based GAN [3], policy based reinforcement learning [4], etc.
>
> 2. The point of Proposition 2 is to show that z-recon loss behaves like a distance between T^{-1}(z) and S(z). So when z-recon is minimized, it implies S gets closer to T^{-1} in the sense of the induced metric. We’ve updated the paper to explicitly state this.
>
> 3. Thanks for the suggestion, but we believe you mean Sec 3.2 (IIUC). We added the pointer there in the new version.
>
> Thank you again for the feedback and interest.
>
> [1] Auto-Encoding Variational Bayes
> [2] Variational Continual Learning
> [3] Calibrating Energy-based Generative Adversarial Networks
> [4] Latent Space Policies for Hierarchical Reinforcement Learning

---

### Official Review · AnonReviewer4 · 2018-11-12
**Interesting ideas but the paper has some issues**

**Rating:** 5
**Confidence:** 5

**Review:**

This paper proposes new methods for distilling a feed-forward generative model (student) from an autoregressive generative model (teacher) as an alternative to the reverse-KL divergence. The first part of the paper analyses optimization issues with the reverse KL divergence while in the second part of the paper alternatives are proposed (x-reconstruction and z-reconstruction).

Detailed comments:

1.
In abstract and other places: "sparse gradient signal from the teacher".
Sparsity implies that many of the values are exactly zero, while Section 3.1 seems to imply that some of the values might be small (or pointing towards the origin).

2.
In Section 3.1 and 3.2 the authors discuss a potential failure mode of the reverse KL:

But, proposition 1 boils down to the fact that if the student's mass is more spread out than the teacher is some direction, that it should shrink that mass closer to zero as well.

In the example of the paper: if an eigenvalue of T is smaller than 1, it would mean that the student which is spherical Gaussian, would adjust its probability mass to also be smaller in that eigenvector's direction.

As training progresses, the students mass would be much closer to the teacher and the probability of 'pointing away' from the origin would be about as likely as pointing towards.

So it's not clear at all that the described property is problematic for optimization, as it could as well be interpreted as the student trying to fit the teacher's distribution better.

3.
Was the KL between P_S(x_i | z_<i) and P_T(x_i | x_<i) computed analytically? If these conditional distributions are Gaussian (which they are in many of the examples) this should be trivial.

4.
Section 4 about the neural vocoder needs to be expanded: many details are missing here and although it's one of the more important experiments in the paper it's relatively neglected compared to the other parts of the paper.

5.
In the Section 4: the experiment with reverse-KL is a straw man comparison: For audio the reverse KL was only proposed in combination with the power loss (Oord et al). Two additional experiments would make the result a lot stronger: KL+power-loss and X-recon+power-loss. Because if the x-recon method does not work well together with the power-loss, its practical applicability seems limited.


The proposed methods are interesting, because they are elegant and seems to work reasonably well on the tasks tried. The first part of the paper about gradient sparsity/orientation needs to be addressed. Section 4 should be expanded and an additional comparison should be made.

I would change my rating if these issues were addressed.

---

> ### Author Response · Authors · 2018-11-22
> **Response to AnonReviewer4**
>
> Thank you for the close reading and detailed feedback! We address the comments below:
>
> 1. Indeed, by “sparsity“, we mean the probability of effective gradient signals that point away from the mode of the teacher is small, the complement of which are the gradient signals that are either zero or point towards the model of the teacher (when this is viewed binarily: point-toward being 0 and point-away being 1, it means signal 1 is sparse). We have tried to make it clearer in the paper by paraphrasing it as the gradient distribution being skewed, or imbalanced over the orientation of push with respect to the origin.
>
> 2. Depending on the chosen family of the student, this might have different effects. (we've updated the paper to include this discussion and better motivate the analysis in Sec3.1)
>     a. If p_S is independent gaussian (mean field assumption), the best student will be extremely concentrated at the mode of the teacher in high dimension. Here the optimality according to the reverse KL will more likely sacrifice the norm of the samples drawn from the student (see Fig1.3 of [1]). Even in this case, the probability of point-away signal will still not be the same as the probability of point-toward signal. They are only equal when weighted by the magnitude (which means the expected gradient is zero at optimality).
>     b. If p_S is multivariate gaussian,
>         (i) One can rotate and rescale both p_T and p_S according to the covariance matrix of p_S, such that the latter once again becomes standard normal. Doing this is to show that our analysis is without loss of generality: the probability of receiving a point-away gradient signal is determined by the “relative” covariance of p_T (after transformation under covariance of p_S) with respect to the standard normal.
>         (ii) As we argue in the paper, as long as there is correlation present in the now transformed p_T, the condition coefficient (eq3) can be extremely small due to the exponential decrease in the volume of hyper-cone (shaded area of Fig1b). The training algorithm can constantly make progress “in expectation”, but since we’re using SGD in practice, the rate at which p_S becomes better now depends on how likely it is to get a point-away signal, assuming p_S fits to the mode of p_T first. The latter assumption and gradient sparsity posing a problem for optimization were also validated by our experiment (Fig 2a).
>
> 3. The Neural Vocoder experiment used closed-form KL and the rest used monte carlo estimate.
>
> 4. We’ve updated our paper to include more experimental details on the neural vocoder experiment. In particular, we have clarified that we use closed form reverse KL proposed in ClariNet. Each of our flow consists of 10 residual dilated convolution block with kernel width of 2 and 64 output channels.
>
> 5. We’ve added one more comparison with reconstruction loss + power loss, which is included in the following link: https://soundcloud.com/inverse-matching/sets/samples-for-inverse-matching
>
> Again, we thank you for your constructive feedback.
>
> [1] Two problems with variational expectation maximisation for time-series models

---

### Meta-Review · Area_Chair1 · 2018-12-14

**Confidence:** 4
**Recommendation:** Reject

**Metareview:**

The paper proposes new methods for optimization of optimization of KL(student_model||teacher_model).

The topic is relevant. The paper also contains interesting ideas and the proposed methods are interesting; they are elegant and seems to work reasonably well on the tasks tried.

However, the reviewers do not all agree that the paper is well written. The reviewers have pointed out several issues that need to be addresses before the paper can be accepted.